# Palaeogenomes of Eurasian straight-tusked elephants challenge the current view of elephant evolution

Matthias Meyer[1]*, Eleftheria Palkopoulou[2], Sina Baleka[3], Mathias Stiller[1], Kirsty E H Penkman[4], Kurt W Alt[5,6], Yasuko Ishida[7], Dietrich Mania[8], Swapan Mallick[2], Tom Meijer[9], Harald Meller[8], Sarah Nagel[1], Birgit Nickel[1], Sven Ostritz[10], Nadin Rohland[2], Karol Schauer[8], Tim Schüler[10], Alfred L Roca[7], David Reich[2,11,12], Beth Shapiro[13], Michael Hofreiter[3]*

[1]Max Planck Institute for Evolutionary Anthropolgy, Leipzig, Germany; [2]Department of Genetics, Harvard Medical School, Boston, United States; [3]Evolutionary Adaptive Genomics, Institute for Biochemistry and Biology, Department for Mathematics and Natural Sciences, University of Potsdam, Potsdam, Germany; [4]Department of Chemistry, University of York, York, United Kingdom; [5]Center of Natural and Cultural History of Man, Danube Private University, Krems-Stein, Austria; [6]Department of Biomedical Engineering and Integrative Prehistory and Archaeological Science, Basel University, Basel, Switzerland; [7]Department of Animal Sciences, University of Illinois at Urbana-Champaign, Urbana, United States; [8]State Office for Heritage Management and Archaeology Saxony-Anhalt with State Museum of Prehistory, Halle, Germany; [9]Naturalis Biodiversity Center, Leiden, Netherlands; [10]Thüringisches Landesamt für Denkmalpflege und Archäologie, Weimar, Germany; [11]Broad Institute of Harvard and MIT, Cambridge, United States; [12]Howard Hughes Medical Institute, Harvard Medical School, Boston, United States; [13]Department of Ecology and Evolutionary Biology, University of California, Santa Cruz, United States

*For correspondence: mmeyer@eva.mpg.de (MM); michael.hofreiter@uni-potsdam.de (MH)

Competing interests: The authors declare that no competing interests exist.

**Abstract** The straight-tusked elephants *Palaeoloxodon* spp. were widespread across Eurasia during the Pleistocene. Phylogenetic reconstructions using morphological traits have grouped them with Asian elephants (*Elephas maximus*), and many paleontologists place *Palaeoloxodon* within *Elephas*. Here, we report the recovery of full mitochondrial genomes from four and partial nuclear genomes from two *P. antiquus* fossils. These fossils were collected at two sites in Germany, Neumark-Nord and Weimar-Ehringsdorf, and likely date to interglacial periods ~120 and ~244 thousand years ago, respectively. Unexpectedly, nuclear and mitochondrial DNA analyses suggest that *P. antiquus* was a close relative of extant African forest elephants (*Loxodonta cyclotis*). Species previously referred to *Palaeoloxodon* are thus most parsimoniously explained as having diverged from the lineage of *Loxodonta*, indicating that *Loxodonta* has not been constrained to Africa. Our results demonstrate that the current picture of elephant evolution is in need of substantial revision.

## Introduction

In the late Miocene in Africa, the last of several major radiations within Proboscidea gave rise to the family Elephantidae, which comprises living elephants and their extinct relatives including mammoths (genus *Mammuthus*) and various dwarf elephant species from Mediterranean islands. The three living

**eLife digest** Understanding how extinct species are related to each other or to their living relatives is often a difficult task. Many extinct species have been identified only from incomplete fragments of some of their bones. However, even if complete skeletons have been found, determining the relationships between species can be tricky because researchers often have to rely solely on the shapes of the bones.

It is sometimes possible to retrieve DNA sequences from fossil bones. This is easier with younger fossils and those that have been recovered from cold environments. Ancient DNA sequences have been retrieved from only a few fossils older than 100,000 years, but such DNA sequences can be tremendously useful in determining how different species are related to each other.

Today there are three living elephant species: the African forest elephant, the African savanna elephant and the Asian elephant. However, there are many extinct elephant species. For example, the European straight-tusked elephant went extinct at least 30,000 years ago, although most of the fossils that have been discovered are at least 100,000 years old. Straight-tusked elephants are generally assumed to be closely related to the Asian elephant, but this conclusion had been based solely on reconstructing skeletons.

Meyer et al. have now obtained DNA sequences from fossils of four straight-tusked elephants ranging from around 120,000 to 240,000 years in age. These sequences were analysed to determine how straight-tusked elephants are related to the three living elephant species and the extinct mammoth, the DNA sequences for which can be found in public databases. The analyses revealed that straight-tusked elephants are in fact most closely related to the African forest elephant, not the Asian elephant as previously thought.

This result completely changes our picture of elephant evolution and suggests that it is extremely difficult to determine elephant relationships based on the shape of their skeleton alone. It also shows that the African elephant lineage was not restricted to the African continent (the place where all elephant lineages originated), but that it also left Africa.

Overall, the results presented by Meyer et al. confirm that DNA sequences are of critical importance for understanding the evolution of animals. Future research should include obtaining DNA sequences from additional extinct elephant species as well as careful re-evaluation of skeletal measurements for reconstructing elephant evolution.

elephant species (the African savanna elephant, *Loxodonta africana*, the African forest elephant, *L. cyclotis* and the Asian elephant, *Elephas maximus*), represent the last remnants of this family and of the formerly much more widely distributed and species-rich order Proboscidea. Apart from mammoths, the elephant genus with the most abundant fossil record in Eurasia is *Palaeoloxodon* (straight-tusked elephants; *Figure 1*), which appears in Eurasia around 0.75 million years ago (Ma) (*Lister, 2016*). Based on morphological analyses, *Palaeoloxodon* is widely accepted as being more closely related to the extant Asian elephant than to mammoths or extant African elephants (*Shoshani et al., 2007*; *Todd, 2010*) and is often subsumed into the genus *Elephas* (*Maglio, 1973*; *Sanders et al., 2010*). Across its range from Western Europe to Japan, *Palaeoloxodon* probably comprised several species (*Shoshani et al., 2007*), and, based on morphological comparisons, all of them are considered to be derived from the African *Palaeoloxodon* (or *Elephas*) *recki* (*Maglio, 1973*; *Saegusa and Gilbert, 2008*), which was the predominant proboscidean lineage in Africa during the Pliocene and Pleistocene but went extinct around 100 thousand years ago (ka) (*Owen-Smith, 2013*). Straight-tusked elephants may have survived in mainland Eurasia until around 35 ka, although the youngest reliably dated remains are from the last interglacial, 115–130 ka (*Stuart, 2005*).

Recent technological progress has pushed back the temporal limit of ancient DNA research, enabling, for example, recovery of a low coverage genome of a ~700,000 year-old horse preserved in permafrost (*Orlando et al., 2013*). For more temperate regions, however, evidence of DNA preservation reaching far beyond the last glacial period is still limited to a single locality, Sima de los Huesos in Spain, where DNA has been recovered from ~430 ka old hominin and bear remains (*Dabney et al., 2013*; *Meyer et al., 2016*). While genetic analyses of the extinct interglacial fauna

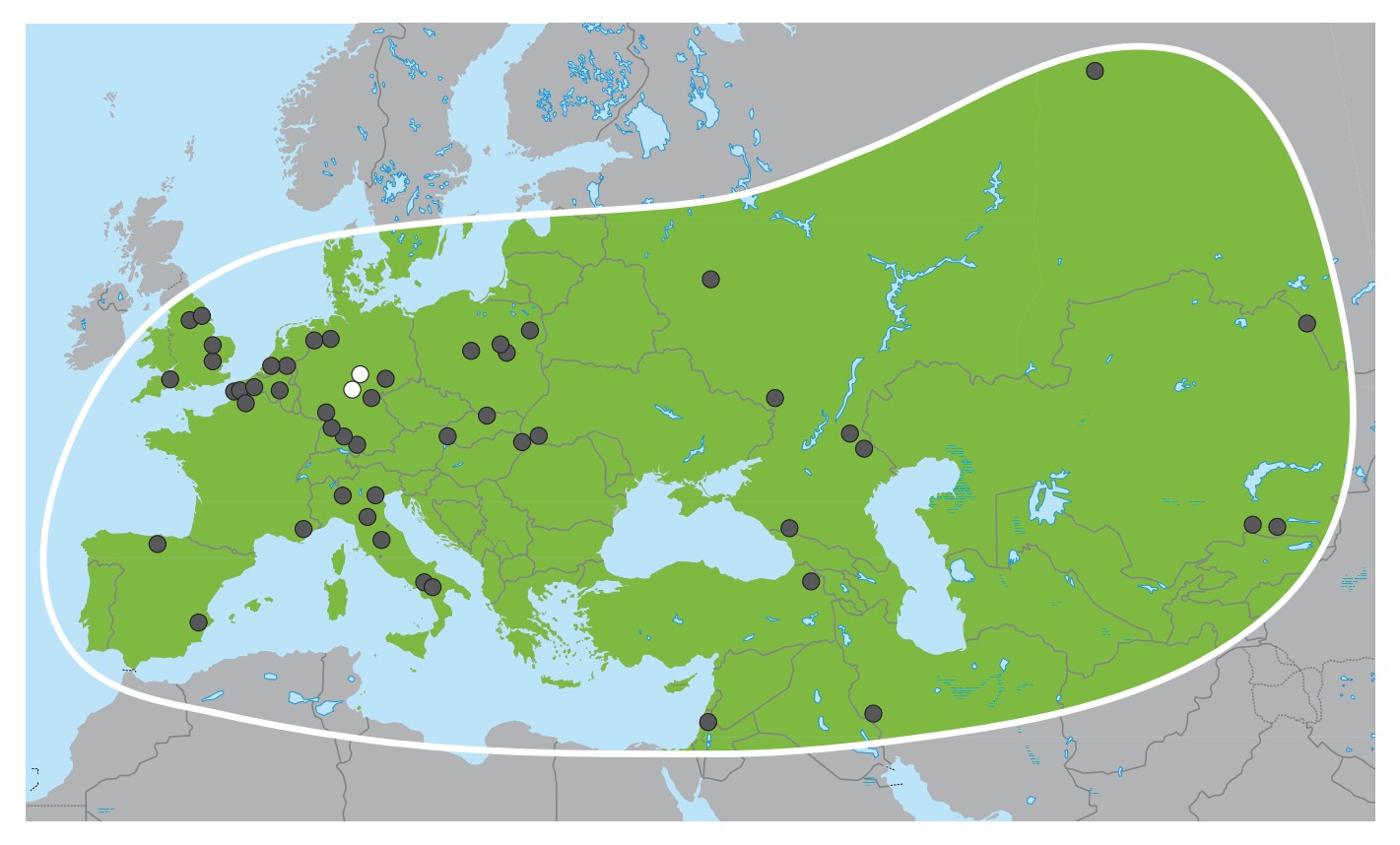

**Figure 1.** *Palaeoloxodon antiquus*, geographic range based on fossil finds (after *Pushkina, 2007*). White dots indicate the locations of Weimar-Ehringsdorf and Neumark-Nord.

remain a challenging undertaking, recent advances in ancient DNA extraction (*Dabney et al., 2013*) and sequencing library construction (*Meyer et al., 2012*) have improved access to highly degraded DNA.

## Results and discussion

To better understand the evolutionary relationships between the extinct straight-tusked elephants and other elephant species, we attempted DNA extraction and sequencing from several *P. antiquus* fossils, four of which we investigated in depth. Three of these, which were all unambiguously assigned to *P. antiquus* based on their morphology, were from Neumark-Nord (NN) 1 in Germany, a fossil-rich site that has been proposed to date to MIS 5e (~120 ka) or MIS 7 (~244 ka) or both (*Mania, 2010*; *Schüler, 2010*; *Penkman, 2010*). This site has yielded one of the largest collections of *P. antiquus* remains known to date. The fourth fossil was recovered during recent active mining in the travertine deposits of Weimar-Ehringsdorf (WE), Germany, a quarry that has for more than a century yielded a rich collection of fossils representing a typical European interglacial fauna (*Kahlke, 1975*). Weimar-Ehringsdorf is best known for the discovery of Neanderthal remains in the early 20th century, and the assemblage is dated to MIS 7 (*Mallick and Frank, 2002*). The *Paleoloxodon* bone fragment from Weimar-Ehringsdorf is morphologically undiagnostic with respect to species. However, it was found in the Lower Travertine, which was dated to ~233 ka (*Schüler, 2003*) and where *P. antiquus* is the only elephantid found so far. We performed DNA extraction, library preparation, hybridization capture and high-throughput sequencing on all four fossils (*Figure 2—source data 1*) and obtained full mitochondrial genome sequences for all of them (*Figure 2—figure supplement 1*). All sequences show short fragment lengths (*Figure 2—figure supplement 2*) and

signals of cytosine deamination compatible with the old age of the specimens (*Figure 2—figure supplement 3*).

We inferred a phylogeny using the four *Paleoloxodon* mitochondrial genomes and mitochondrial genomes from 16 *M. primigenius*, 2 *E. maximus* and 13 *Loxodonta* individuals. The latter were chosen for a diversity of haplotypes, including forest elephant derived ('F-clade') haplotypes as well as 'S-clade' haplotypes found only among savanna elephants (*Debruyne, 2005*). For calibration, we used an estimated divergence of the African elephant lineage from that of Asian elephants and mammoths of 6.6–8.6 Ma (*Rohland et al., 2007*). Surprisingly, *P. antiquus* did not cluster with *E. maximus*, as hypothesized from morphological analyses. Instead, it fell within the mito-genetic diversity of extant *L. cyclotis*, with very high statistical support (*Figure 2*). The four straight-tusked elephants did not cluster together within this mitochondrial clade, but formed two separate lineages that share a common ancestor with an extant *L. cyclotis* lineage 0.7–1.6 Ma (NN) and 1.5–3.0 Ma (WE) ago, respectively. However, mitochondrial DNA represents a single, maternally inherited locus and does not reflect the full evolutionary history of populations or species. Furthermore, the transfer of mitochondrial DNA between hybridizing species is not unusual when gene flow is strongly male-mediated (*Petit and Excoffier, 2009*; *Li et al., 2016*; *Cahill et al., 2013*), as is the case with elephants. For example, mitochondrial sequences of the F-clade have also been found in some *L. africana* individuals (*Debruyne, 2005*) despite the very substantial divergence of their nuclear genomes (*Roca et al., 2005*; *Rohland et al., 2010*), a pattern that has been attributed to mitochondrial gene flow from forest to savanna elephants (*Roca et al., 2005*).

We therefore performed shotgun sequencing of DNA libraries prepared from the two best-preserved NN individuals (a petrous bone and a molar) to recover nuclear DNA sequences. When mapped to the *L. africana* reference genome, 39% and 28% of the sequence reads generated from these specimens were identified as elephant, respectively (*Figure 2—source data 1*). A neighbor-joining phylogenetic tree based on ~770M (petrous) and ~210M (molar) base pairs of *P. antiquus* nuclear DNA placed *P. antiquus* and *L. cyclotis* as sister taxa to the exclusion of *L. africana* (*Figure 2*). A tree with identical topology was obtained using coding sequences only and a maximum likelihood approach (*Figure 2—figure supplement 4*). Despite the high sequence error rates associated with the low-coverage genomes generated from the *P. antiquus* specimens, all nodes in the nuclear trees show maximal bootstrap support. The mitochondrial and nuclear phylogenies thus support a sister group relationship between *P. antiquus* and *L. cyclotis*.

Despite their geographical proximity, the WE and NN specimens are found in different positions in the mitochondrial tree. Given that the three NN specimens show highly similar mitogenome sequences, we considered whether the sites date to different interglacials. Electron Spin Resonance (ESR) dating of tooth enamel has been applied to both sites, suggesting an age of ~117 ka (range 97–142 ka [*Schüler, 2010*]) for the NN1 layers from which our samples originate and of 233 ka (range 216–250 ka [*Schüler, 2003*]) for the WE specimen. In order to better estimate the age of the NN1 site, we used amino acid racemization of snail opercula (*Penkman et al., 2011*), including samples from the continental Eemian type-site of Amersfoort (*Zagwijn, 1961*) (*Figure 2—source data 1*), which is correlated with MIS 5e (*Cleveringa et al., 2000*). The NN1 opercula show similar (perhaps slightly lower) levels of biomolecular degradation compared to Amersfoort, suggesting an Eemian age for NN1 (*Figure 2—figure supplement 5*). Importantly, NN1 shows very similar levels of amino acid racemization in intra-crystalline protein decomposition as a second site at Neumark-Nord (NN2), *indicating* that both sites are of the same age. Considerable evidence supports an Eemian age for NN2, including palaeomagnetic data that shows a correlation with the MIS 5e Blake event and thermoluminescence dating of flint to ~126 ± 6 ka (*Sier et al., 2011*). These results therefore indicate an Eemian age also for NN1. Since the WE specimen likely dates to the previous interglacial, this suggests that the very different mitogenomes between WE and NN1 may reflect the contraction and re-expansion of the range of *P. antiquus* across glacial cycles.

Our results have implications both for understanding elephant evolutionary history and for the use of morphological data to decipher phylogenetic relationships among elephants. The strongly supported mitochondrial and nuclear DNA phylogenies clearly demonstrate that *Palaeoloxodon antiquus* is more closely related to *Loxodonta* than to *Elephas* (*Figure 3*), suggesting that *Elephas antiquus* should not be used synonymously for *Paleoloxodon antiquus* when referring to the taxon. The new phylogeny suggests a remarkable degree of evolutionary transformation, from an ancestor that possessed the features of the cranium and dentition of *Loxodonta*, shared by both *L. africana* and *L.*

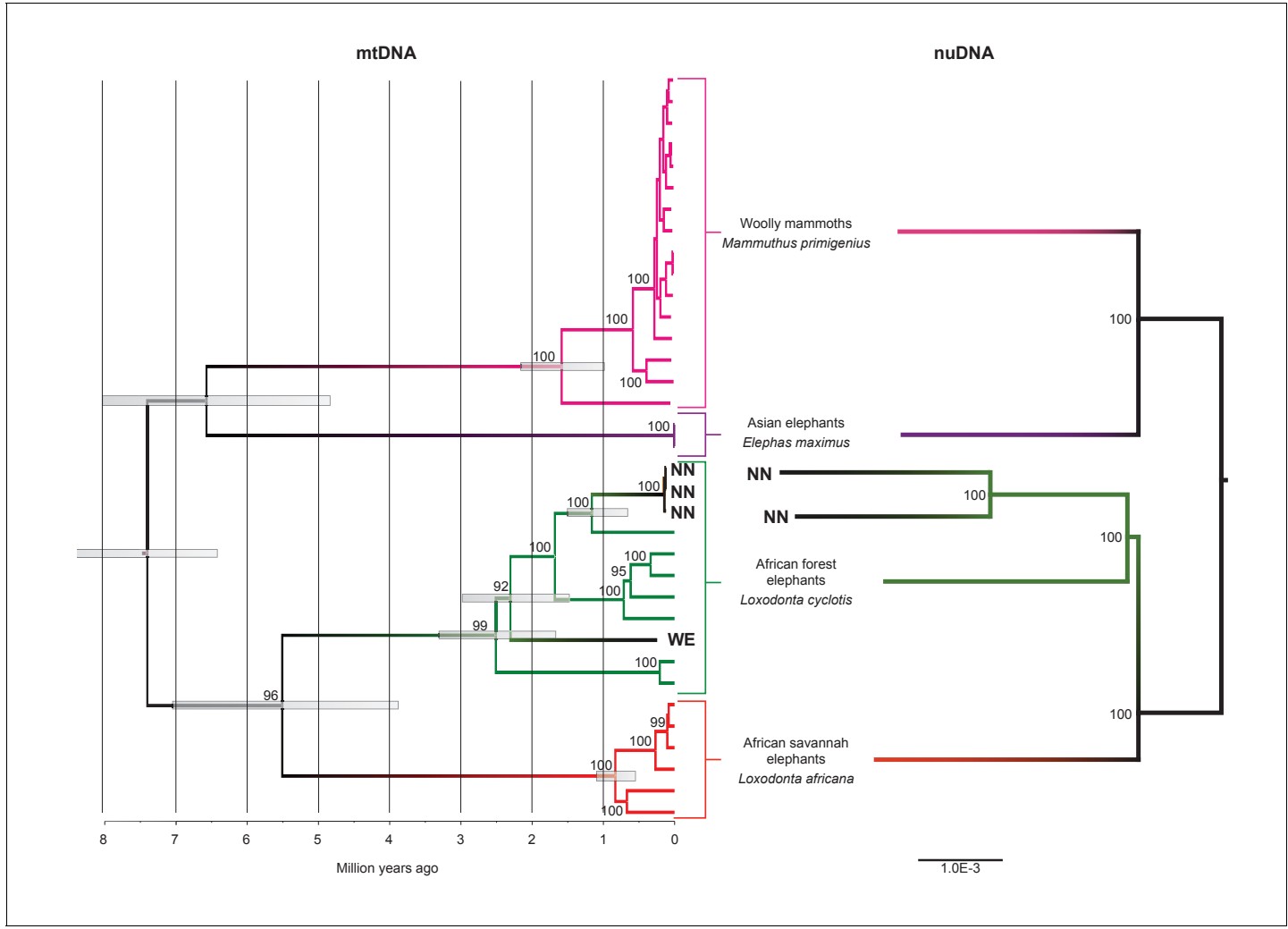

**Figure 2.** Phylogenetic trees relating the mitochondrial and nuclear sequences of *P.antiquus* (NN and WE) to other elephantids. (A) Maximum clade credibility (MCC) tree resulting from a BEAST (*Drummond et al., 2012*) analysis of 35 complete mitochondrial genomes using 15,447 sites. Node bars and numbers show the 95% highest posterior density estimates for node ages and clade support, respectively. Mitochondrial partitioning scheme and molecular and coalescent models are described in 'Materials and methods'. (B) Pairwise-distance Neighbor-joining tree from between 210 million and 2.5 billion base pairs of nuclear shotgun sequence data. Bootstrap support values from 100 replicates are shown inside nodes. Summary statistics of the underlying sequence data are available in *Figure 2—source data 1*.

The following source data and figure supplements are available for figure 2:

**Source data 1.** This spreadsheet contains summary statistics of all sequence data generated in this study, the sequences of PCR primers used for reconstructing mtDNA sequences of extant elephants, as well as amino acid racemization data on opercula of *Bithynia tentaculata* from Amersfoort.

**Figure supplement 1.** Sequence coverage of the NN and WE mitochondrial genomes.

**Figure supplement 2.** DNA fragment size distribution inferred from full-length mtDNA sequences.

**Figure supplement 3.** Frequency of C to T substitutions for each position in the sequence alignments.

**Figure supplement 4.** Maximum likelihood tree from concatenated nuclear protein-coding sequences with bootstrap support values shown inside nodes.

**Figure supplement 5.** Amino acid racemization data.

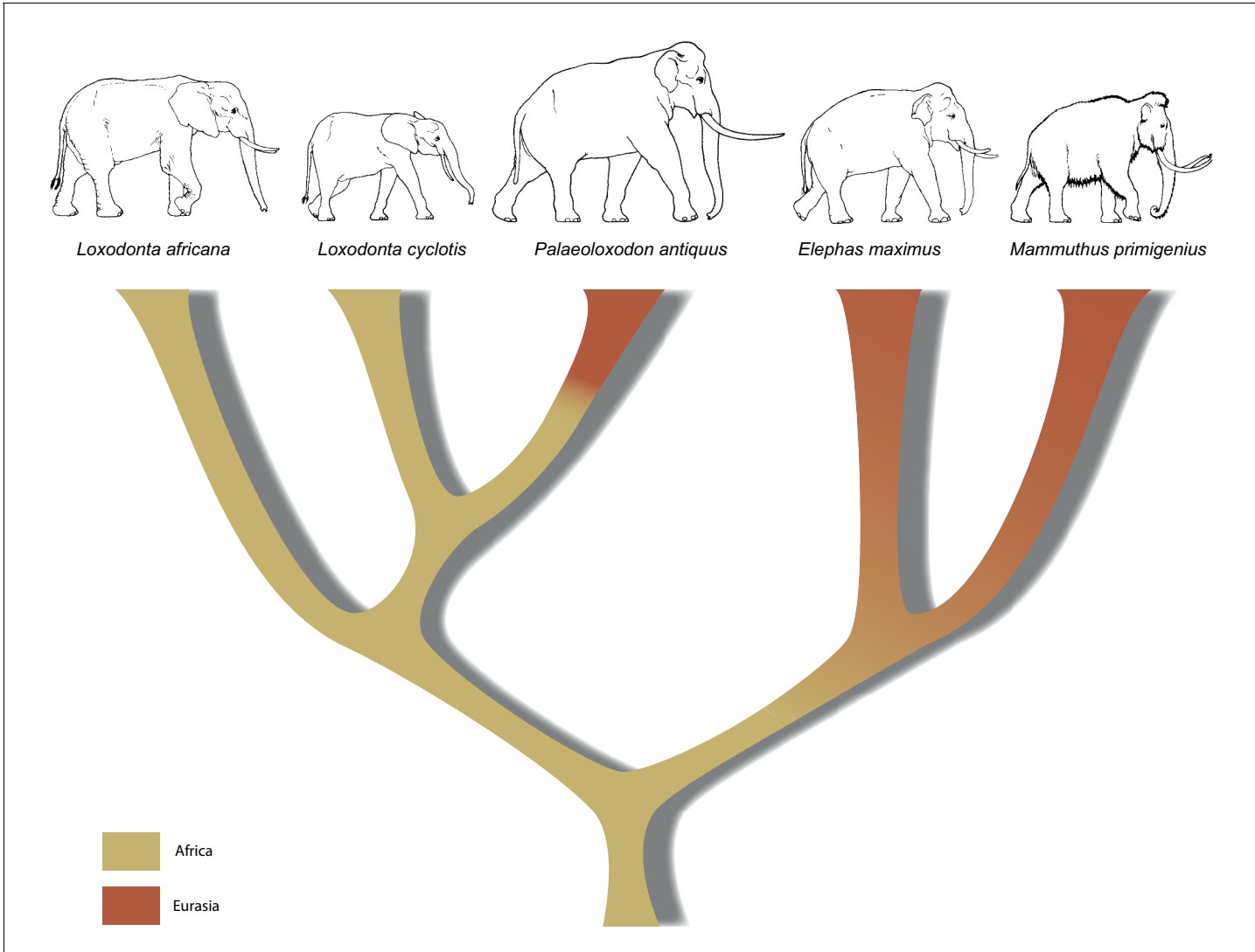

**Figure 3.** A revised tree of phylogenetic relationships among elephantids, color-coded by their presumed geographical range.

*cyclotis* (*Maglio, 1973*; *Sanders et al., 2010*), to a descendant that is highly similar to *Elephas* (sensu stricto, the lineage of the Asian elephant) in many morphological features. However, it should be noted that currently available genomic data from elephantids only allow for reconstructing the broad picture of elephant evolution. More complex evolutionary scenarios are conceivable, which might explain the presence of some *Elephas*-like traits in *P. antiquus*. These could for example involve gene flow, as has been shown for *L. africana* and *L. cyclotis* based on mitochondrial evidence (*Roca et al., 2005*). In addition, the very large effective population size of the forest elephants (*Rohland et al., 2010*) could have allowed the retention of ancestral traits by incomplete lineage sorting.

In summary, the molecular results presented here urge for a re-examination of morphology across the Elephantidae. This is especially important as the fossil record for elephants dates back several million years, well beyond the survival of ancient DNA. If, for example, *P. recki*, which was the most abundant Pleistocene elephant species in Africa, is indeed ancestral to *P. antiquus* and thus also represents a member of the *Loxodonta* lineage, the interpretation of the fossil record of elephantids in Africa is in strong need of revision. Furthermore, in contrast to the genera *Mammuthus* and *Elephas*, which also had their origin in Africa, the lineage of *Loxodonta* is generally assumed never to have left Africa. Although *Osborn, 1942* placed *Palaeoloxodon* in the Loxodontinae on the basis of

several cranial characters, later authors (*Shoshani et al., 2007*; *Todd, 2010*) have rejected this placement in favor of a placement in Elephantinae, restricting *Loxodonta* (and Loxodontinae) geographically to Africa. However, our data reveal that the *Loxodonta* lineage (as *Paleoloxodon*) also colonized the Eurasian continent. Last, the finding that *L. africana* is genetically more distant from *L. cyclotis* than is *P. antiquus* strongly supports previous evidence that urged recognition of *L. cyclotis* and *L. africana* as distinct species and underlines the importance of conservation efforts directed toward African forest elephants.

## Material and methods

### Sampling, DNA extraction and library preparation

In January 2014, a fragment of an elephant long bone was discovered during work at the Ehringsdorf quarries. The specimen was removed from the lower travertine (~3m above the base and 2.5 m below the Pariser horizon), which has been dated by micro probe U/Th-series dating of primary travertine (*Mallick and Frank, 2002*) and ESR dating of tooth enamel (*Schüler, 2003*) to ~233 ka. A piece of the bone (inventory number 14/18–1) was transferred to the ancient DNA laboratory at the MPI-EVA in Leipzig. Ten extracts were prepared using between 36 and 53 mg of bone (totaling 425 mg) material following the method of Dabney *et al.* 2013 (*Dabney et al., 2013*). From these extracts, 30 libraries were prepared using single-stranded library preparation (*Gansauge and Meyer, 2013*) with input volumes of 4, 8 and 12 µl DNA extract (of 25 µl extract volume), respectively. The number of library molecules was determined by digital droplet PCR using Bio-Rad's (Hercules, CA) QX200 system with EvaGreen chemistry (QX200 ddPCR EvaGreen Supermix, Bio-Rad) and primers IS7 and IS8 (*Meyer and Kircher, 2010*) following the manufacturer's instructions (*Figure 2—source data 1*). Libraries were then amplified using AccuPrime Pfx DNA polymerase (Thermo Fisher Scientific, Waltham, MA) (*Dabney and Meyer, 2012*) and labeled with two sample-specific indices (*Kircher et al., 2012*).

Ancient DNA work on the Neumark-Nord specimens was carried out in the ancient DNA laboratory at the University of Potsdam. Initially, eight specimens were screened for the presence of elephant DNA of which the two best preserved ones (individual 23, Landesmuseum Halle museum inventory number HK 2007:25.285,117; a molar fragment [NEU2A] and a fragmentary upper jaw [NEU8B]; inventory number HK 92:990) were selected for further analyses. In addition, in 2013, the petrous bone of individual 30 (NEPEC; inventory number HK 2007:25:280 = E15.1.96) was sampled and also used in the analysis. Fourteen DNA extracts were prepared from individual 30, and six from each of the two other specimens, using approximately 50 mg of bone powder in each extraction. DNA extraction and library preparation were performed as described above but including *Archaeoglobus fulgidus* uracil-DNA glycosylase in library preparation (*Gansauge and Meyer, 2013*), which removes the majority of uracils that are typically present in ancient DNA fragments. In addition, to maximize yields in library preparation, two extracts (25 µl each) from each specimen were combined and 40 µl were used as input for library preparation. Reaction volumes in steps 1–3 of the protocol were doubled to accommodate larger input volumes of extract. Optimal amplification cycle numbers were established using qPCR (PikoReal Real-Time PCR system, Thermo Fisher Scientific) with primers IS7 and IS8 (*Gansauge and Meyer, 2013*). Libraries were then amplified and labeled with one sample-specific index. After purification (MinElute PCR purification kit, Qiagen, Germany), the different libraries for each sample were pooled.

### Enrichment and sequencing of mtDNA

52-mer capture probes for the enrichment of mtDNA sequences from elephants were designed using the published mtDNA genome sequences of African forest elephant (NC_020759), Asian elephant (NC_005129), African savanna elephant (NC_000934) and the mastodon (NC_009574), with one probe starting at each position in these genomes. Probes containing simple repeats longer than 24 bp (repetition of the same 1–8 bp sequence motif) were removed. Single-stranded biotinylated DNA probes were generated as described elsewhere (*Fu et al., 2013*) and used for two successive rounds of hybridization capture following a bead-based protocol (*Maricic et al., 2010*). Enriched libraries were pooled and sequenced on one lane of a HiSeq2500 (Illumina, San Diego, CA) in

paired-end mode (2 × 76 cycles plus 2 × 7 cycles index reads; Weimar-Ehringsdorf libraries) or on an Illumina NextSeq 500 (2 × 76 cycles plus 1 × 8 cycles index read; Neumark-Nord libraries).

## Mitochondrial sequence data processing and consensus calling

Sequences were assigned to their source library requiring perfect matches to one of the expected indices or index pairs and overlap-merged to reconstruct full-length molecule sequences (*Renaud et al., 2014*). Due to the different properties of the data obtained from Weimar-Ehrings-dorf and Neumark-Nord with regard to sequence length distribution and damage patterns (*Figure 2—figure supplements 2* and *3*), two different strategies were used for mapping and consensus calling. To minimize the loss of alignments due to the high frequencies of damaged-induced substitutions in the Weimar-Ehringsdorf data, mapping to the *L. cyclotis* mtDNA genome (JN673264) was performed as previously described for the Sima de los Huesos mtDNA assemblies (*Dabney et al., 2013*), using BWA and allowing up to five C to T substitutions but not more than three of other types. The sequences from Neumark-Nord were mapped with 'ancient' parameters as described elsewhere (*Meyer et al., 2012*). PCR duplicates were removed with bam-rmdup (*Stenzel, 2014* Bio-hazard, available from https://bitbucket.org/ustenzel/biohazard) by calling a consensus from sequences with identical alignment start and end coordinates. Sequences shorter than 30 bp were discarded. An overview of the DNA extracts, libraries and sequences generated in this study is provided in *Figure 2—source data 1*.

When visually inspecting the Weimar-Ehringsdorf sequence alignments, we identified several regions in the mitochondrial genome where more than one sequence variant was present. Based on BLAST searches on a subset of these sequences, we found that they derived from present-day human or microbial contamination. We thus aligned all sequences to the identified contaminant genomes (GenBank accession nos. NC_012920, AF365635 and CP008889) and removed sequences that showed a greater similarity to one of the contaminants than to the African forest elephant mtDNA. No removal of contaminant sequences was necessary for the Neumark-Nord samples. To minimize the impact of damage-derived C to T substitutions on consensus calling, all T occurring in the first and last three positions of the Weimar-Ehringsdorf sequences were substituted by N. Next, a position-based tabular output was generated from the alignment files using the 'mpileup' function of SAMtools (*Li et al., 2009*). This file was used to call the consensus at positions with minimum sequence coverage of 3 if the sequences were in at least 67.0% agreement. At three positions in the mtDNA genome (positions 384, 8467, 8469) with low consensus support, we spotted obvious alignment errors in one or all specimens and determined the consensus base manually. Apart from a ~500 bp stretch of repetitive sequence in the D-loop, which cannot be reconstructed with short DNA fragments, only four positions remain undetermined in the Weimar-Ehringsdorf sequence and even fewer (between none and three) in the Neumark-Nord sequences.

## MtDNA phylogenetic reconstructions

We estimated mitochondrial phylogenies using the software BEAST (*Drummond et al., 2012*) v 1.8.2 and a data set including 31 complete mitochondrial genomes (GenBank accession nos.; *L. cyclotis*: JN673264, JN673263, KJ557424, KJ557423, KY616976, KY616979, KY616978; *L. africana*: WA4020, KR0014, KR0138, NC000934, DQ316069, AB443879; *E. maximus*: NC005129, DQ316068; *M. primigenius*: DQ316067, NC007596, EU155210, EU153449, EU153455, EU153456, EU153458, EU153445, EU153446, EU153447, EU153448, EU153452, EU153453, EU153454, JF912200; *M. columbi*: JF912199). For three of the *L. cyclotis* individuals (LO3505, LO3508 and DS1511) and three *L. africana* individuals (WA4020, KR0014, KR0138), only partial mitochondrial sequences were previously published. Full genome sequences were obtained using previously collected samples (*Ishida et al., 2013*) and the amplification and sequencing strategy detailed by *Brandt et al. (2012)*, except that additional primers were used in sequencing (*Figure 2—source data 1*). The complete mitochondrial genome sequences were partitioned prior to analysis into four partitions, representing concatenated genes (with *ND6* reversed), tRNAs, rRNAs, and the control region, and analyses were performed with and without the control region fragment. All BEAST analyses were performed assuming the flexible skygrid coalescent model (*Gill et al., 2013*) and the uncorrelated lognormal relaxed molecular clock (*Drummond et al., 2006*). We calibrated the molecular clock using the ages of ancient tips and a lognormal prior with a mean of 7.6 million years and standard deviation of

500,000 years for the divergence of the *Loxodonta* and *Elephas/Mammuthus* lineages (*Rohland et al., 2007*). Ages of the ancient samples were sampled from normal distributions derived from stratigraphic and previously estimated radiometric dates: Neumark-Nord: 142–92 ka (*Schüler, 2010*); Ehringsdorf 250–216 ka (*Mallick and Frank, 2002*). Separate evolutionary rates and models of nucleotide substitution, as estimated using jModelTest (*Posada, 2008*), were estimated for each partition in the alignment. We ran two MCMC chains for 60 million iterations each, with trees and model parameters sampled every 6000 iterations. Chain convergence and parameter sampling were examined by eye using Tracer v 1.6 (Rambaut A, Suchard MA, Xie D & Drummond AJ (2014) Tracer v1.6, available from http://beast.bio.ed.ac.uk/Tracer). The first 10% of samples were discarded from each run after which the two runs were combined. Trees were summarized and maximum clade credibility (MCC) trees identified using TreeAnnotator v 1.8.2, which is distributed as part of the BEAST package. MCC trees were edited and annotated using FigTree v1.4.2 (http://tree.bio.ed.ac.uk/software/figtree/).

## Shotgun sequencing of nuclear DNA

Libraries from the three Neumark-Nord samples and another sample (not included in this study) were pooled in equimolar concentrations and shotgun-sequenced on an Illumina NextSeq 500 (2 × 76 bp cycles) at Harvard Medical School. Following determination of endogenous content and complexity in each library, two of them (NEPEC from the petrous bone and NEU2A from the molar fragment) were chosen for additional sequencing and were pooled together with another sample (not included in this study) for one NextSeq 500 run.

## Nuclear sequence data processing

Sequences were assigned to their source library according to their index allowing for one mismatch. Adapters were trimmed and paired-end sequences were merged with SeqPrep 1.1 (https://github.com/jstjohn/SeqPrep) using default parameters but with a modification in the source code to retain the best base quality scores in the merged region. Merged sequences shorter than 30 bp were discarded. Alignment to the African savanna elephant reference genome (loxAfr4; downloaded from ftp://ftp.broadinstitute.org/distribution/assemblies/mammals/elephant/loxAfr4/) was performed with BWA's version 0.7.8 (*Li and Durbin, 2009*) using 'ancient' parameters and SAMtools' v.0.1.19 'samse' command (*Li et al., 2009*). A custom script was used to remove duplicates, which takes into account the alignment coordinates of both ends of the sorted sequences and their orientation.

From the first sequencing run, 47% and 35% of the sequences from the libraries from the petrous bone and the molar fragment (NEPEC and NEU2A, respectively) aligned to the reference genome while only 0.5% of the sequences from the third library (NEU8B) aligned to the reference genome. The high percentage of mapped sequences in the petrous sample is consistent with previous reports on the superior DNA preservation in this part of the skeleton (*Gamba et al., 2014*). Following the second sequencing run, the total endogenous content of the first two libraries was estimated to 39% and 28%, with an average sequence length of 39 bp and 38 bp, respectively (*Figure 2—source data 1*). The average depth of coverage was 0.65-fold for NEPEC and 0.14-fold for NEU2A. Both of them showed low frequencies of C to T substitutions at the 5' and 3' end, which are characteristic for *Afu* UDG-treated single-stranded libraries (*Gansauge and Meyer, 2013*), except for in CpG context, where deamination of 5-methylcytosine leaves thymine and not uracil (*Figure 2—figure supplement 3*).

We also processed sequencing data from an African forest elephant (SL0001) that was sequenced to high-coverage at the Broad Institute and re-processed sequencing data of an Asian elephant from (*Lynch et al., 2015*). We trimmed adapters with SeqPrep 1.1 using default parameters and aligned paired-end reads to loxAfr4 using BWA's 'aln' algorithm and SAMtools' 'sampe' command. Duplicate reads were removed with SAMtools's 'rmdup'. Moreover, we used the high-coverage genome of a woolly mammoth (Wrangel) from (*Palkopoulou et al., 2015*). The woolly mammoth alignments were re-processed for removal of duplicate reads with the custom script mentioned above.

## Nuclear DNA phylogenetic reconstruction

To determine the phylogenetic relationships between the two *P. antiquus* specimens and other members of the *Elephantidae* family, we called pseudo-haploid consensus sequences for all

autosomes of the two *P. antiquus* samples (~770 and~210 million sites, respectively). Sites with base quality below 30 and reads with mapping quality below 30 were filtered out. To exclude post-mortem damage-derived C to T substitutions, we trimmed 2 bp from the ends of all reads. We included regions of the loxAfr4 genome for which at least 90% of all possible 35-mers do not find a match at another position allowing for up to one mismatch, similar to the mappability filter described in (*Prüfer et al., 2014*). We used a majority-allele calling rule that required at least one read aligned at each position of the genome. Using the same approach, we called sequences for an Asian elephant (Uno [*Lynch et al., 2015*]), a woolly mammoth (Wrangel [*Palkopoulou et al., 2015*]) and an African forest elephant (SL0001; Broad Institute). We also used the reference sequence loxAfr4, as an African savanna elephant. We estimated the number of differences per base-pair for pairwise comparisons of all sequences and constructed a distance matrix, from which we built a Neighbor-joining (NJ) tree using PHYLIP version 3.696 (*Felsenstein, 2005*). To obtain support values for the nodes of the tree, we performed a bootstrap analysis (100 replicates) by splitting all autosomes in blocks of 5 Mb and randomly sampling blocks with replacement and built a majority-rule consensus tree.

We also extracted coding DNA sites (CDS) of protein-coding genes using the Ensembl 87 release for the loxAfr3 genome (downloaded from http://www.ensembl.org/) from each elephant genome sequence. CDS mapping to unknown chromosomes as well as CDS containing partial codons were excluded, resulting in a total of 86,212 CDS. Multiple sequence alignments were generated for each gene using MAFFT –ginsi (*Katoh et al., 2002*; *Katoh and Standley, 2013*) with 1000 iterations, which were concatenated into a single fasta file. Maximum likelihood phylogenetic analysis was performed with RAxML v8.2 (*Stamatakis, 2014*) with the GTRGAMMA model of nucleotide substitutions and 100 bootstrap trees. The resulting phylogeny is identical in its topology to that of the NJ tree with 100% bootstrap support (*Figure 2—figure supplement 4*).

## Dating the specimens

Amino acid racemization (AAR) analyses were undertaken on the intra-crystalline protein from four individual *Bithynia tentaculata* opercula from the Eemian type-site, Amersfoort (*Cleveringa et al., 2000*): Amersfoort-1, upper depth 27.71, lower depth 28.50 (NEaar 2982–3, 3972 and 4681) and compared with previously published data from a single horizon at Neumark-Nord 1 (15.5.87/2, Schluffmudde, 25 cm under Anmoor = surface of the lower shore area; NEaar 5698–5703 [*Penkman, 2010*]) and several horizons from Neumark-Nord 2 (*Sier et al., 2011*). All samples were prepared using procedures of isolating the intra-crystalline protein by bleaching (*Penkman et al., 2008*). Two subsamples were then taken from each shell; one fraction was directly demineralized and the free amino acids analyzed (referred to as the 'free' amino acids, FAA, F), and the second was treated to release the peptide-bound amino acids, thus yielding the 'total' amino acid concentration, referred to as the 'total hydrolysable amino acid fraction (THAA, H*). Samples were analyzed in duplicate by RP-HPLC. During preparative hydrolysis, both asparagine and glutamine undergo rapid irreversible deamination to aspartic acid and glutamic acid, respectively (*Hill, 1965*). It is therefore not possible to distinguish between the acidic amino acids and their derivatives and they are reported together as Asx and Glx, respectively. The D/L values of aspartic acid/asparagine, glutamic acid/glutamine, alanine and valine (D/L Asx, Glx, Ala, Val) are then assessed to provide an overall estimate of intra-crystalline protein decomposition (*Penkman et al., 2011*).

## Acknowledgements

We thank the Elephant Genome Sequencing Consortium, especially Elinor Karlsson and Jessica Alfoldi, for permission to use the African forest elephant genome data. We thank Chloé Piot, Ronny Barr and Susanne Sicker for help with preparing the figures, Adam Brandt and Tolulope Perrin-Stowe for sequence information and Adrian Lister for comments on the manuscript. Mitochondrial genome sequences were deposited in GenBank under accession nos. KY499555 - KY499558, KY616976, KY616978 and KY616979, nuclear sequence alignments are available through the European Nucleotide Archive (ENA) project no. PRJEB18563.

MM is supported by the Max Planck Society. BS was supported by the Gordon and Betty Moore Foundation. AR and YI were supported by the US Fish and Wildlife Service. The amino acid analyses were supported by Wellcome Trust and Leverhulme Trust funding to KP, with thanks to Sheila Taylor

for technical support. D.R. is an investigator of the Howard Hughes Medical Institute. MH is supported by ERC consolidator grant 310763 GeneFlow.

## Additional information

### Funding

| Funder | Grant reference number | Author |
|---|---|---|
| Max Planck Society | Open-access funding | Matthias Meyer<br>Mathias Stiller<br>Sarah Nagel |
| Gordon and Betty Moore Foundation | | Beth Shapiro |
| US Fish and Wildlife Service | AFE1606-F16AP00909 | Yasuko Ishida<br>Alfred L. Roca |
| Wellcome Trust | | Kirsty E. H. Penkman |
| Leverhulme Trust | | Kirsty E. H. Penkman |
| European Research Council | 310763 GeneFlow | Michael Hofreiter |

The funders had no role in study design, data collection and interpretation, or the decision to submit the work for publication.

### Author contributions

MM, Conceptualization, Data curation, Formal analysis, Supervision, Validation, Investigation, Visualization, Methodology, Writing—original draft, Project administration, Writing—review and editing; EP, Data curation, Formal analysis, Validation, Investigation, Methodology, Writing—original draft, Writing—review and editing; SB, Conceptualization, Data curation, Validation, Investigation, Methodology, Writing—review and editing; MS, Formal analysis, Validation, Investigation, Writing—review and editing; KEHP, Resources, Formal analysis, Funding acquisition, Validation, Investigation, Methodology, Writing—original draft, Writing—review and editing; KWA, DM, TM, HM, SO, Resources, Investigation, Writing—review and editing; YI, Resources, Data curation, Formal analysis, Validation, Investigation, Writing—review and editing; SM, Data curation, Validation, Writing—review and editing; SN, BN, NR, Investigation, Methodology, Writing—review and editing; KS, Investigation, Visualization, Writing—review and editing; TS, Conceptualization, Resources, Investigation, Writing—review and editing; ALR, Conceptualization, Resources, Data curation, Supervision, Validation, Project administration, Writing—review and editing; DR, Conceptualization, Data curation, Supervision, Validation, Investigation, Methodology, Writing—review and editing; BS, Conceptualization, Data curation, Formal analysis, Validation, Investigation, Methodology, Writing—review and editing; MH, Conceptualization, Data curation, Formal analysis, Supervision, Validation, Investigation, Visualization, Writing—original draft, Project administration, Writing—review and editing

### Author ORCIDs

Matthias Meyer, http://orcid.org/0000-0002-4760-558X
Kirsty E H Penkman, http://orcid.org/0000-0002-6226-9799

## Additional files

### Major datasets

The following datasets were generated:

| Author(s) | Year | Dataset title | Dataset URL | Database, license, and accessibility information |
|---|---|---|---|---|
| Palkopoulou E, Baleka S, Mallick S, | 2017 | Nuclear DNA sequences from two Palaeoloxodon antiquus fossils | http://www.ebi.ac.uk/ena/data/search?query= | Publicly available at EBI European |

| | | | | | |
|---|---|---|---|---|---|
| Rohland N, Reich D, Hofreiter M | | | | PRJEB18563 | Nucleotide Archive (accession no: PRJEB18563) |
| Ishida I, Roca AL | 2017 | Mitochondrial DNA sequences from 3 Loxodonta cyclotis individuals | | https://www.ncbi.nlm. nih.gov/nuccore/ KY616976 | Publicly available at NCBI GenBank (accession no: KY616976) |
| Ishida I, Roca AL | 2017 | Mitochondrial DNA sequences from 3 Loxodonta cyclotis individuals | | https://www.ncbi.nlm. nih.gov/nuccore/ KY616978 | Publicly available at NCBI GenBank (accession no: KY616978) |
| Ishida I, Roca AL | 2017 | Mitochondrial DNA sequences from 3 Loxodonta cyclotis individuals | | https://www.ncbi.nlm. nih.gov/nuccore/ KY616979 | Publicly available at NCBI GenBank (accession no: KY616979) |
| Meyer M, Baleka S, Stiller M, Nagel S, Nickel B, Schüler T, Hofreiter M | 2017 | Mitochondrial DNA sequences from 4 Palaeoloxodon antiquus fossils | | https://www.ncbi.nlm. nih.gov/nuccore/ KY499555 | Publicly available at NCBI GenBank (accession no: KY499555) |
| Meyer M, Baleka S, Stiller M, Nagel S, Nickel B, Schüler T, Hofreiter M | 2017 | Mitochondrial DNA sequences from 4 Palaeoloxodon antiquus fossils | | https://www.ncbi.nlm. nih.gov/nuccore/ KY499556 | Publicly available at NCBI GenBank (accession no: KY499556) |
| Meyer M, Baleka S, Stiller M, Nagel S, Nickel B, Schüler T, Hofreiter M | 2017 | Mitochondrial DNA sequences from 4 Palaeoloxodon antiquus fossils | | https://www.ncbi.nlm. nih.gov/nuccore/ KY499557 | Publicly available at NCBI GenBank (accession no: KY499557) |
| Meyer M, Baleka S, Stiller M, Nagel S, Nickel B, Schüler T, Hofreiter M | 2017 | Mitochondrial DNA sequences from 4 Palaeoloxodon antiquus fossils | | https://www.ncbi.nlm. nih.gov/nuccore/ KY499558 | Publicly available at NCBI GenBank (accession no: KY499558) |

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
