## [Decision Letter]

Thank you for submitting your article "Palaeogenomes of Eurasian straight-tusked elephants challenge the current view of elephant evolution" for consideration by *eLife*. Your article has been favorably evaluated by Diethard Tautz (Senior Editor) and three reviewers, one of whom is a member of our Board of Reviewing Editors. The following individuals involved in review of your submission have agreed to reveal their identity: William Sanders (Reviewer #2); Thomas Gilbert (Reviewer #3).

The reviewers have discussed the reviews with one another and the Reviewing Editor has drafted this decision to help you prepare a revised submission.

Summary:

The compact manuscript by Meyer and colleagues reports the sequencing of four full mitochondrial genomes and two partial nuclear genomes from *Palaeoloxodon antiquus* fossils. *P. antiquus* is a species that is thought to have been quite widespread throughout Eurasia in the Pleistocene. Phylogenetic analyses of these mitochondrial and nuclear data yield the surprising conclusion that the closest extant relatives of *P. antiquus* are not the Asian elephants (genus *Elephas*), as suggested by morphology, but the African elephants (genus *Loxodonta*). If the view offered by the genetic data holds, the current interpretation of the evolution of elephant morphology would need to be substantially revised. In addition, the fossils from which ancient DNA was extracted are quite old (~120-244 thousand years ago), so it is remarkable that such old samples yielded so much data.

Essential revisions:

1) It is not true that no one has suggested that relatives of *Loxodonta* made it out of Africa; Kretzoi did so emphatically in 1950, and others (e.g., van den Bergh; Markov and Saegusa) have accepted the possibility without endorsing the notion. These studies should be discussed and cited in the revised manuscript.

2) The authors do not treat the paleontological/morphological hypothesis about the relationships of *P. antiquus/Loxodonta/Elephas* as a competing hypothesis – they dismiss it in a single sentence, saying morphological features showing *P. antiquus* to be closely related to *Elephas* (such as *E. recki*) are convergences. And then go on to act as if elephant systematics were in a shamble and need serious revision – this simply isn't the case. They should present their view as a competing hypothesis and say if further work supports their hypothesis, then it has serious implications for elephant systematics and here is what those implications are. But, morphology isn't as labile as the authors suggest, and they do not provide compelling selective reasons for why a forest elephant such as *P. antiquus* would converge morphologically on a savanna grazer such as *E. recki*.

For example, one good test would be to study the abundant postcrania of *Loxodonta, P. antiquus*, and *E. recki* – there are clear differences in many aspects of the postcranium, especially in manus and pes elements, between *Loxodonta* and *Elephas* – see where *Palaeoloxodon* falls out. Additionally, it will be essential for the authors to discuss how additional species that were not sampled could be critical in resolving the proposed conundrum between the morphological and genetic results. Examples of such species include *Palaeoloxodon namadicus*, a straight-tusked species from Asia, *Elephas planifrons* and *E. hysudricus* (and other fossil species of *Elephas*, such as *E. playcephalus*) or any fossil species of *Loxodonta*, such as the abundant material from Laetoli of *L. exoptata*. Finally, it would be interesting to know if the authors tried but failed to sample any of the *Elephas ekorensis* – *E. recki – E. iolensis* lineage.

3) Execution of the phylogenetic analyses. The authors used two different methods on the mtDNA (Bayesian inference) and nuclear (neighbor-joining) data. It is not clear to me why that was the case. Bayesian inference is quite well established. In contrast, neighbor-joining is great for generating quick-and-dirty trees during the data exploration phase of experiments, but not very good in exploring tree space (the model of sequence evolution employed is also not described). I would recommend the authors also use BEAST to analyze the nuclear data (and at the same time estimate divergence times on that data set as well). If the use of BEAST is problematic due to computational constraints (if that's the case, the authors should include a brief description of those constraints in the manuscript), then my suggestion would be to use maximum likelihood to infer the nuclear phylogeny (e.g., use RAxML with the PROTGAMMAAUTOF option) and then r8s to calculate the divergence times.

4) Interpretation of the phylogenetic results, which are shown in Figure 2. The branch lengths of the African forest elephant – *Palaeoloxodon* clade are rather different between the mitochondrial and the nuclear phylogenetic trees. For example, the three NN samples are nearly identical in the mtDNA phylogeny but quite divergent in the nuclear one even though in general the nuclear genes appear to be somewhat slower evolving than the mitochondrial ones. In contrast, the key internode supporting the monophyly of the African forest elephant-Palaeoloxodon clade is quite long in the mtDNA phylogeny but very short in the nuclear phylogeny. Please discuss these two "discrepancies" and provide potential explanations.

---

## [Author Response]

*Essential revisions:*

*1) It is not true that no one has suggested that relatives of Loxodonta made it out of Africa; Kretzoi did so emphatically in 1950, and others (e.g., van den Bergh; Markov and Saegusa) have accepted the possibility without endorsing the notion. These studies should be discussed and cited in the revised manuscript.*

We could not access Kretzoi 1950, but we were able to obtain copies of Markov and Saegusa 2008 and van der Bergh 1999. Both publications discuss Kretzoi 1950. However, van der Bergh concludes that *Stegoloxodon* represents "a dwarfed member of the Elephantinae" whereas Markov and Saegusa do not discuss the phylogenetic affiliation at all, but restrict their discussion to which taxonomic name should correctly be used for the sparse material available. Nevertheless, to acknowledge that an out-of-Africa migration of Loxodontinae has been proposed previously, even though this idea has been rejected by more recent cladistic studies, we have added a sentence to the Discussion: "Although Osborn (1942) placed *Palaeoloxodon* in the Loxodontinae on the basis of several cranial characters, later authors (Shoshani et al. 2007, Todd 2010) have rejected this placement in favour of a placement in Elephantinae, restricting (and Loxodontinae) geographically to Africa”. We also slightly modified the Abstract and the main text where we are discussing this issue.

2) The authors do not treat the paleontological/morphological hypothesis about the relationships of P. antiquus/Loxodonta/Elephas as a competing hypothesis – they dismiss it in a single sentence, saying morphological features showing P. antiquus to be closely related to Elephas (such as E. recki) are convergences. And then go on to act as if elephant systematics were in a shamble and need serious revision – this simply isn't the case. They should present their view as a competing hypothesis and say if further work supports their hypothesis, then it has serious implications for elephant systematics and here is what those implications are. But, morphology isn't as labile as the authors suggest, and they do not provide compelling selective reasons for why a forest elephant such as P. antiquus would converge morphologically on a savanna grazer such as E. recki.

*For example, one good test would be to study the abundant postcrania of Loxodonta, P. antiquus, and E. recki – there are clear differences in many aspects of the postcranium, especially in manus and pes elements, between Loxodonta and Elephas – see where Palaeoloxodon falls out. Additionally, it will be essential for the authors to discuss how additional species that were not sampled could be critical in resolving the proposed conundrum between the morphological and genetic results. Examples of such species include Palaeoloxodon namadicus, a straight-tusked species from Asia, Elephas planifrons and E. hysudricus (and other fossil species of Elephas, such as E. playcephalus) or any fossil species of Loxodonta, such as the abundant material from Laetoli of L. exoptata. Finally, it would be interesting to know if the authors tried but failed to sample any of the Elephas ekorensis – E. recki – E. iolensis lineage.*

The genetic evidence for the sister group relationship between *P. antiquus* and *Loxodonta cyclotis* is based on millions of mostly neutrally evolving markers. For this reason we do not agree that the morphological and genetic analyses should be given equal weight in the discussion of elephant systematics. We are however more cautious in our conclusions now. In addition to emphasizing that “[…] the currently available genomic data from elephantids only allow for reconstructing the broad picture of elephant evolution” we now state that “More complex evolutionary scenarios are conceivable, which might explain the presence of some *Elephas*-like traits in *P. antiquus*. These could for example involve gene flow, as has been shown for *L. africana* and *L. cyclotis* based on mitochondrial evidence (Roca, Georgiadis, and O'Brien 2005)”. We now also mention the possibility that “[…] the very large effective population size of the forest elephants (Rohland et al. 2010) could have allowed the retention of ancestral traits by incomplete lineage sorting.”

*As in the previous version of the text, we remain cautious in our speculations about the phylogenetic position of P. recki: “If, for example, P. recki, which was the most abundant Pleistocene elephant species in Africa, is indeed ancestral to P. antiquus and thus also represents a member of the Loxodonta lineage, the interpretation of the fossil record of elephantids in Africa is in strong need of revision”. However, we have further toned down our conclusions in several places to acknowledge the concerns by the reviewer. While we agree that additional morphological and DNA sequence data would be interesting, the former is not our expertise (and also not feasible within a reasonable timeframe) and the latter is likely not possible as most fossil species mentioned exceed the age limits of ancient DNA preservation. We have now made this clear in Discussion: “[…] the fossil record for elephants dates back several million years, well beyond the survival of ancient DNA.”.*

*3) Execution of the phylogenetic analyses. The authors used two different methods on the mtDNA (Bayesian inference) and nuclear (neighbor-joining) data. It is not clear to me why that was the case. Bayesian inference is quite well established. In contrast, neighbor-joining is great for generating quick-and-dirty trees during the data exploration phase of experiments, but not very good in exploring tree space (the model of sequence evolution employed is also not described). I would recommend the authors also use BEAST to analyze the nuclear data (and at the same time estimate divergence times on that data set as well). If the use of BEAST is problematic due to computational constraints (if that's the case, the authors should include a brief description of those constraints in the manuscript), then my suggestion would be to use maximum likelihood to infer the nuclear phylogeny (e.g., use RAxML with the PROTGAMMAAUTOF option) and then r8s to calculate the divergence times.*

Bayesian inference is indeed well established for phylogenetic analysis of single or few loci, such as mitochondrial DNA or unlinked noncoding loci, but is computationally not feasible for complete nuclear genome sequences. Apart from computational constraints, BEAST is also not appropriate for the analysis of nuclear genomes since it implements a model that assumes no recombination within loci. For these reasons, in the previous version of the manuscript we chose to construct a distance-based phylogeny using the neighbor-joining algorithm, which is practical and fast for analyzing large datasets and thus the most popular method for complete nuclear genomes. Given the amount of information in our nuclear dataset, NJ is expected to reconstruct the true tree with high confidence (100% bootstrap support). However, in the revised version of the manuscript, we followed the reviewers’ suggestion to use a maximum likelihood approach and have added a RAxML phylogenetic analysis of concatenated nuclear coding sequences from our genome-wide data (did not use the complete nuclear genomes due to computational constraints). The resulting topology is identical to that of the NJ tree, again with 100% bootstrap support. We added a description of this analysis to the Methods section and a brief statement to Results: “A tree with identical topology was obtained using coding sequences only and a maximum likelihood approach (Figure 2—figure supplement 7).” We refrain from estimating divergence times from the nuclear data since high coverage genomic data from all taxa, including *P. antiquus* would be required to obtain robust estimates. Errors in the *P. antiquus* sequence data, which is ancient and of relatively low quality (< 1x depth of coverage) could produce artifacts.

*4) Interpretation of the phylogenetic results, which are shown in Figure 2. The branch lengths of the African forest elephant – Palaeoloxodon clade are rather different between the mitochondrial and the nuclear phylogenetic trees. For example, the three NN samples are nearly identical in the mtDNA phylogeny but quite divergent in the nuclear one even though in general the nuclear genes appear to be somewhat slower evolving than the mitochondrial ones. In contrast, the key internode supporting the monophyly of the African forest elephant-Palaeoloxodon clade is quite long in the mtDNA phylogeny but very short in the nuclear phylogeny. Please discuss these two "discrepancies" and provide potential explanations.*

Branch lengths in the mitochondrial and nuclear trees are not expected to be the same. The mitochondrial tree depicts the coalescence of a single lineage while the nuclear tree depicts the coalescence of tens of thousands of nuclear loci across the genome. Hence the difference in the branch lengths of the NN samples. In addition, whereas the mitochondrial sequences are of high quality, sequencing error and DNA damage in the low coverage genomes contribute private mutations to the respective branches of the nuclear tree. We have added a statement to point to this issue: “Despite the high sequence error rates associated with the low coverage genomes generated from the *P. antiquus* specimens, all nodes in the nuclear trees show maximal bootstrap support.”

Regarding the node supporting the monophyly of the African forest elephant-*Paleoloxodon* clade, the mitochondrial tree suggests that forest elephants and our *P. antiquus* samples carry relatively similar mitochondrial genomes but are rather diverged in their nuclear genomes (as suggested by the nuclear phylogeny). This is not unusual in elephants, for instance as mentioned in the manuscript, some savanna elephants carry forest-type mtDNA although their nuclear genomes are deeply diverged.